# Reinforcement Mid-Training

**Yijun Tian**[1,†]**, Shaoyu Chen**[2,†]**, Zhichao Xu**[3]**, Yawei Wang**[4]**, Jinhe Bi**[5]**, Peng Han**[1]**, Wei Wang**[1,*]
[1] Xi'an Jiaotong University [2] Shanghai Jiao Tong University [3] University of Utah
[4] The George Washington University [5] Ludwig Maximilian University of Munich
meetyijun@gmail.com        wei.wang@xjtu.edu.cn

## Abstract

The development of state-of-the-art large language models is commonly understood as a two-stage process involving pre-training and post-training. We point out the need for an additional intermediate stage called reinforcement mid-training with potential for strong performance gains. In this paper, we formally define the problem and identify three key challenges: (1) inefficient training due to excessive reasoning steps, (2) disregard of the imbalanced token entropy distribution, and (3) underutilization of token information. To address these challenges, we propose RMT, a framework for efficient, adaptive, and unified reinforcement mid-training with various innovative components. In particular, we first introduce a dynamic token budget mechanism that constrains unnecessary reasoning steps and mitigates model overthinking. Next, we design a curriculum-based adaptive sampling method that fosters a progressive learning trajectory from easy to hard tokens. Finally, we present a dual training strategy that combines reinforcement learning with next-token prediction, ensuring targeted learning on key tokens and full exploitation of all token information. Extensive experiments demonstrate the superiority of RMT over state-of-the-art methods, achieving up to **+64.91%** performance improvement with only **21%** of the reasoning length in language modeling. We also show that checkpoints obtained after reinforcement mid-training can benefit the subsequent post-training, yielding up to **+18.76%** improvement in the mathematical domain.

## 1 Introduction

The development of state-of-the-art large language models is commonly understood as a two-stage process (Minaee et al., 2024; Kumar et al., 2025; Tian et al., 2025). The first stage, pre-training, involves training a model on vast, unlabeled web-scale data using objectives such as next-token prediction. This stage equips the model with broad world knowledge and linguistic competence (Devlin et al., 2019a; Raffel et al., 2020; Brown et al., 2020). The second stage, post-training, aims to align the model with human objectives or downstream tasks, often using specialized, high-quality datasets. At this stage, the goal is to instill capabilities such as instruction following, tool use, agentic reasoning, and preference alignment, typically through methods like supervised fine-tuning (Wang et al., 2022; Chung et al., 2024) and reinforcement learning (Ouyang et al., 2022; Lai et al., 2025).

However, this two-stage view overlooks a crucial, emerging intermediate phase: mid-training, which leverages the large-scale unlabeled pre-training data with more targeted objectives than general pre-training, systematically enhancing complex capabilities like mathematical reasoning to better prepare the model for subsequent post-training (Wang et al., 2025b; OLMo et al., 2025; Wake et al., 2024). Unlike post-training that usually depends on domain-specific data with labels or reward signals, mid-training relies on pre-training data that does not require extra human annotations or verified rewards. The foundational concept of applying reinforcement learning to the mid-training stage was pioneered in the Reinforcement Pre-Training approach (Dong et al., 2025). Despite its name, this method technically functions in a mid-training stage. The base model they used has already demonstrated established instruction-following and reasoning capabilities, positioning it clearly beyond the pre-training stage. However, existing work faces three critical challenges:

---

[†]Equal Contribution    [*]Corresponding Author

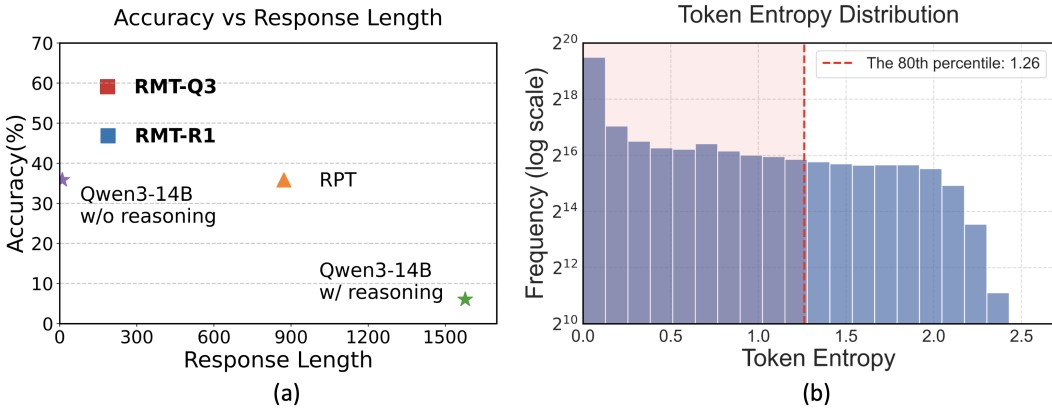

Figure 1: (a) Our proposed models RMT-Q3 and RMT-R1 achieves superior performance with significant less reasoning length. (b) The majority of the tokens are of low-entropy.

- **Inefficient Reinforcement Learning with Overthinking**: The thinking process is unconstrained and leads the model to generate excessively long reasoning chains, causing additional computation overhead during training and inference. For example, in Figure 1 (a), existing method requires significantly more tokens for reasoning while cannot guarantees a satisfactory performance.

- **Overlook of Imbalanced Token Entropy Distribution**: Tokens vary in entropy, reflecting various levels of uncertainty and learning difficulty. Existing approaches sample challenging tokens indiscriminately, which can overload the model with highly difficult tokens during the early stages of training, before it has sufficient capacity.

- **Underutilization of Token Information**: As shown in Figure 1 (b), the majority of tokens in the data exhibit low entropy. Existing method only support the training on high-entropy tokens, neglecting the vast majority of low-entropy tokens. For unlabeled pre-training data, where every token contributes to language understanding, this exclusion leads to substantial information loss and missed learning opportunities.

In this paper, we first formally define the problem of Reinforcement Mid-Training. To solve this problem and overcome the challenges, we propose RMT, a **R**einforced **M**id-**T**raining framework that is efficient, adaptive, and unified. RMT contains several innovative designs: (1) To address the overthinking problem, we introduce a dynamic token budget mechanism that adaptively constrains the length of the reasoning process, encouraging the model to become more concise and efficient. (2) By taking into account the imbalanced token entropy distribution, we present a curriculum-based adaptive sampling method that guides the model on a progressive learning trajectory from easy to hard tokens. This ensures stable learning while gradually increasing the difficulty to master complex reasoning. (3) To incorporate every token information, we propose a dual training strategy that synergistically combines token-selective reinforcement learning with token-inclusive next-token prediction. This ensures that the model benefits from targeted, reward-driven learning on challenging tokens while still capturing the full information from all other tokens. Language modeling experiments show that RMT can achieve up to **+64.91%** performance improvement with only **21%** of the reasoning length. We further demonstrate the checkpoints obtained from reinforcement mid-training can benefit later post-training stage, with up to **+18.76%** performance improvement. Moreover, our method's plug-and-play nature makes it effective with different backbone model choices and reinforcement learning algorithms. To summarize, our major contributions in this paper are as follows:

- We formally define the problem of Reinforcement Mid-Training and point out three critical challenges: inefficient overthinking, imbalanced token entropy distribution, and the necessity of utilizing all token information.

- To address the challenges, we propose RMT, a plug-and-play framework that are efficient, adaptive, and unified. RMT contains several innovative designs, including a dynamic token budget mechanism, a curriculum-based adaptive sampling method, and a dual training strategy.

- Extensive experiments on both language modeling and continual post-training demonstrate the superiority of RMT over state-of-the-art methods in both effectiveness and efficiency, establishing a stronger foundation for post-training stage.

## 2 RELATED WORK

**LLM Pre-Training.** Language models are predominantly trained with unsupervised pre-training, a process that equips the model with both linguistic fluency and broad world knowledge. Early works include BERT (Devlin et al., 2019b), T5 (Raffel et al., 2020), and GPT-2 (Radford et al., 2019). Modern LLMs like GPT-3 (Brown et al., 2020) establish the dominant paradigm of causal language modeling via next-token prediction. Scaling law research (Kaplan et al., 2020; Hoffmann et al., 2022) further elucidated how compute, data, and parameters interact to enhance model pre-training performance, laying the groundwork for recent advanced models such as LLaMA (Touvron et al., 2023), GPT-4 (Achiam et al., 2023), and Gemini (Gemini, 2023; Comanici et al., 2025).

**LLM Post-Training.** Post-Training refers to the techniques and methodologies employed after a model has undergone pre-training, aiming to refine and adapt the model for specific tasks or user requirements (Tie et al., 2025; Du et al., 2025). From a methodology perspective, existing post-training methods mainly leverage supervised fine-tuning (SFT) (Wang et al., 2022; Wei et al., 2022; Chung et al., 2024) and reinforcement learning (RL) (Ziegler et al., 2019; Rafailov et al., 2023; Wang et al., 2025a; Li et al., 2026). Recently, RL has gained significant attention from industry and academia due to its effectiveness. While RLHF (Ouyang et al., 2022; Bai et al., 2022) with PPO algorithm (Schulman et al., 2017) remains the dominant framework, recent works have been proposed to tackle the critical limitations in stability and computational efficiency of PPO. Notable advances include GRPO (Shao et al., 2024), DAPO (Yu et al., 2025), VAPO (Yue et al., 2025), and GFPO (Shrivastava et al., 2025). Curriculum learning (Hammoud et al., 2025; Chen et al., 2025) and the combination of SFT and RL have also been introduced to further expanded the RL paradigm.

**LLM Mid-Training.** Compared to the well-studied paradigms of pre-training and post-training, mid-training has only recently emerged as a distinct stage in the LLM lifecycle, positioned between the two. Following the definition of (Wang et al., 2025b), mid-training can be seen as an intermediate phase in both computational and data demands, designed to preparing models for post-training. While prior efforts (Wake et al., 2024; OLMo et al., 2025; Wang et al., 2025b) have explored these directions, the role of reinforcement learning (RL) in this stage remains underdeveloped. The sole attempt so far, RPT (Dong et al., 2025) does not resolve the central challenge of bringing RL into mid-training, i.e., its prohibitive computational cost over massive unlabeled corpora. In this work, we formalize the problem of Reinforcement Mid-Training and introduce an efficient framework that enables RL to be applied effectively at this stage.

## 3 REINFORCEMENT MID-TRAINING

In this section, we formally define Reinforcement Mid-Training and present the problem definition.

**What is Reinforcement Mid-Training?** The model is optimized using reinforcement learning on large-scale unlabeled pre-training data, with the goal of enhancing subsequent post-training performance on downstream tasks. In this framework, the model is encouraged to allocate reinforcement learning updates primarily to key tokens that are the most influential for reasoning, while relying on standard next-token prediction for the remaining tokens. The central challenge is to reduce the computational cost of reinforcement learning while ensuring concise and efficient reasoning.

**Problem Definition.** *Given a sequence of tokens $\mathcal{S} = \{\tau_1, \tau_2, ..., \tau_L\}$ of length $L$, we partition the tokens into two disjoint subsets: $\Phi_{RL} \subset \mathcal{S}$ denotes the tokens selected for reinforcement learning and $\Phi_{NTP} = \mathcal{S} \setminus \Phi_{RL}$ represents the tokens for standard next-token prediction, with $|\Phi_{NTP}| \gg |\Phi_{RL}|$. Given a token $\tau_m \in \Phi_{RL}$ with position $m \leq L$ for reinforcement learning, the policy model $\pi_\theta$ takes the preceding tokens $\{\tau_{<m}\} \subset \mathcal{S}$ as input and predicts the next token $\tau_m$ by sampling response $o_m = (c_m, y_m)$, where $c_m$ denotes chain-of-thought reasoning and $y_m$ is the predicted token. A matching reward $r$ is introduced by comparing the exact match of prediction $y_m$ and the ground truth $\tau_m$. A reinforcement learning algorithm is applied to optimize the model with the rewards obtained. In the meantime, for every token $\tau_n \in \Phi_{NTP}$, we perform the next-token prediction by maximizing the likelihood $\log \pi_\theta(\tau_n \mid \tau_{<n})$.*

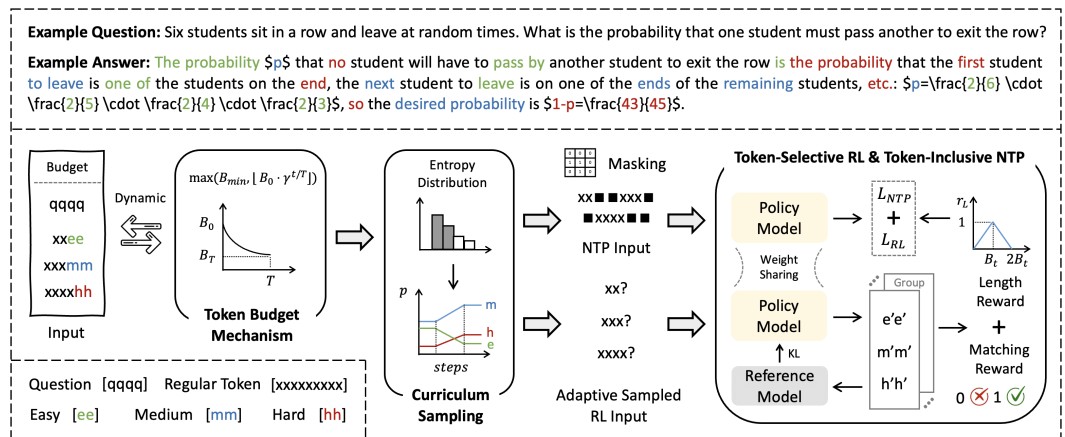

Figure 2: The overall framework. We first identify the difficulty of each token using entropy as a measure of uncertainty. Then, a token budget is assigned to dynamically control the generation length. Building on this, we introduce curriculum-based sampling to encourage the model learning more from easier tokens during early stages and challenging tokens as its capabilities strengthen. Next, we perform reinforcement learning on sampled tokens with length rewards and verifiable matching rewards to enforce training efficiency and accuracy. At the same time, the majority low-entropy tokens are incorporated through next-token prediction with a unified training objective.

## 4 METHOD

In this section, we formally present RMT to address the challenges described in Introduction. RMT contains three innovative components, including a dynamic token budget mechanism, a curriculum-based adaptive sampling strategy, and a unified training objective of token-selective reinforcement learning and token-inclusive next-token prediction. Figure 2 illustrates the framework of RMT.

### 4.1 DYNAMIC TOKEN BUDGET MECHANISM

To mitigate the overthinking problem and enable efficient reasoning, we introduce a token budget to dynamically constrains reasoning length while maintaining prediction quality, through a tailored decay factor coupled with length-based reward signals. In particular, let $T$ denote the total number of training steps and $t \in \{0, 1, \ldots, T\}$ represent the current training step, we define a training step-dependent token budget $B_t$ that decays exponentially with training progress:

$$B_t = \max\left(B_{\min}, \left\lfloor B_0 \cdot \gamma^{t/T} \right\rfloor\right), \tag{1}$$

where $B_{\min} \geq 1$ is the minimum allowable budget to ensure non-trivial reasoning capacity is preserved, $B_0$ is the initial budget, and $\gamma \in (0, 1)$ is the decay factor controlling the rate of budget reduction. At training step $t$, we inject a budget-aware instruction into the input prompt, specifically requesting the model to utilize exactly $B_t$ tokens within the designated `<think></think>` reasoning segment. To further encourage the model to adhere to the target budget $B_t$ while maintaining flexibility, we design a length reward function $r_{\text{len}}(\ell; B_t)$ with a triangular profile centered at the target budget. Let $\ell$ denote the actual number of tokens generated in the reasoning trace. The length reward function is defined as follows:

$$r_{\text{len}}(\ell; B_t) = \begin{cases} r_{\max} \cdot \frac{\ell}{B_t}, & \text{if } 0 \leq \ell \leq B_t, \\ r_{\max} \cdot \frac{2B_t - \ell}{B_t}, & \text{if } B_t < \ell \leq 2B_t, \\ 0, & \text{if } \ell > 2B_t, \end{cases} \tag{2}$$

where $r_{\max}$ denotes the maximum reward and the obtained length reward $r_{\text{len}}$ is used in later training stage. This reward structure exhibits several desirable properties: (1) it achieves its maximum value $r_{\max}$ when $\ell = B_t$, encouraging precise budget adherence, (2) it linearly penalizes both insufficient reasoning ($\ell < B_t$) and excessive verbosity ($B_t < \ell \leq 2B_t$), and (3) it provides zero reward

for severely over-budget reasoning ($\ell > 2B_t$), effectively discouraging computational waste. The integration of this dynamic budget mechanism enables the model to adaptively reduce reasoning overhead as its capabilities mature, leading to more efficient training dynamics and improved resource utilization without sacrificing performance.

## 4.2 CURRICULUM-BASED ADAPTIVE SAMPLING

To effectively exploit the inherent imbalance in token entropy distribution and support adaptive model training, we design a curriculum-based sampling method that prioritizes tokens according to their difficulty levels. This enforces the model to learn from easier tokens in early stages, while gradually shift the attention to challenging tokens as the model's capabilities strengthen. Specifically, we categorize tokens into various difficulty levels based on their entropy values, categorized as `easy`, `medium`, and `hard`. During training, tokens are sampled from these categories with dynamically evolving probabilities. Formally, let $T$ denotes the total number of steps. We define two curriculum transition points $t_1$ and $t_2$ for progressive adjustment, where $0 < t_1 < t_2 < T$. The corresponding sampling probabilities $\mathbf{p}^{t_1}$ and $\mathbf{p}^{t_2}$ over difficulty categories are defined as:

$$\mathbf{p}^{t_1} = \left(p_{\text{easy}}^{t_1}, p_{\text{medium}}^{t_1}, p_{\text{hard}}^{t_1}\right), \quad \mathbf{p}^{t_2} = \left(p_{\text{easy}}^{t_2}, p_{\text{medium}}^{t_2}, p_{\text{hard}}^{t_2}\right), \tag{3}$$

where we constraint $\sum_d p_d^{(\cdot)} = 1$ across different difficulty level $d$ to ensure valid distributions. To enable an adaptive learning, the sampling probability $\mathbf{p}^t$ is changing according to a curriculum schedule, which is implemented via a piecewise linear interpolation that smoothly transitions the sampling probabilities over difficulty levels:

$$\mathbf{p}^t = \begin{cases} \mathbf{p}^{t_1}, & \text{if } t < t_1, \\ (1 - \tau)\mathbf{p}^{t_1} + \tau \mathbf{p}^{t_2}, & \text{if } t_1 \le t < t_2, \text{ where } \tau = \frac{t - t_1}{t_2 - t_1}, \\ \mathbf{p}^{t_2}, & \text{if } t \ge t_2, \end{cases} \tag{4}$$

where $t$ is the current training step and $\tau$ is the decaying factor. The adaptive sampling process operates in two stages. We first sample a difficulty level $d \sim \text{Categorical}(\mathbf{p}^t)$ according to the current curriculum probability. Then, we select uniformly among available candidates at that difficulty level to form a token set $\mathcal{C}$ for later training.

This entropy-driven curriculum-based sampling strategy separates the training into three distinct phases: (1) Early training ($t < t_1$) emphasizes `easy` and `medium` difficulty tokens to establish stable convergence on simpler reasoning patterns. (2) Transition phase ($t_1 \le t < t_2$) provides smooth interpolation that gradually increases exposure to `medium` and `hard` entropy tokens as reasoning capabilities strengthen. (3) Late training ($t \ge t_2$) maintains a stable sampling probabilities that leveraging challenging high-entropy `hard` tokens to boost the model performance.

## 4.3 TOKEN-SELECTIVE RL AND TOKEN-INCLUSIVE NTP

To address the token information underutilization problem and ensure the contribution of all tokens, we present a unified training objective that synergistically combines token-selective reinforcement learning with token-inclusive next-token prediction. These two objectives perform different functions. Reinforcement learning are performed on a small set of selected tokens with different levels of difficulties, while next-token prediction is leveraged to enable the model learn from the majority of low-entropy tokens.

**Token-Selective Reinforcement Learning.** Given the selected token set $\mathcal{C}$, a token $\tau \in \mathcal{C}$, and the token position $pos$ in the sentence, we ask the model to reason about the next token conditioned on the prefix context $\tau_{<pos}$. Specifically, the policy model $\pi_\theta$ takes $\{\tau_{<pos}\}$ as input and predicts the next token $\tau_{pos}$ by sampling $G$ responses $\{o_{pos}^i\}_{i=1}^G$. Each response $o_{pos}^i = (c_{pos}^i, y_{pos}^i)$ consists of a chain-of-thought reasoning sequence $c_{pos}^i$ and a final prediction sequence $y_{pos}^i$. We employ a verifiable matching reward $r_{pos}^i$ for $i$-th response and position $pos$ by comparing the exact match of prediction $y_{pos}^i$ and ground truth next token $\tau_{pos}$. We then form a per-sample composite reward $r^i$ by adding $r_{pos}^i$ and the dynamic length reward $r_{\text{len}}^i$ obtained from Eq. 2. The procedure is as follows:

$$r_{pos}^i = \begin{cases} 1, & \text{if } y_{pos}^i = \tau_{pos}, \\ 0, & \text{otherwise}, \end{cases}; \quad r^i = (1 - w) \cdot r_{pos}^i + w \cdot r_{\text{len}}^i, \tag{5}$$

where $w \in [0, 1]$ is the trade-off weight balancing both rewards, ensuring both accuracy and conciseness. To obtain a stable learning signal, we employ GRPO as our reinforcement learning algorithm and compute a group-relative, whitened advantage from $\{r^i\}_{i=1}^{G}$:

$$\bar{r} \;=\; \frac{1}{G} \sum_{i=1}^{G} r^i, \qquad \sigma \;=\; \mathrm{Std}\big(\{r^i\}_{i=1}^{G}\big), \qquad A^i \;=\; \frac{r^i - \bar{r}}{\sigma + \delta}, \tag{6}$$

where $\bar{r}$ is the mean reward across all $G$ samples, $\sigma$ is the standard deviation of the rewards, and $\delta > 0$ is a small constant to prevent division by zero. The normalized advantage $A^i$ measures how much the $i$-th sample's reward deviates from the group average, thereby guiding the policy update toward relatively better-performing outputs. Finally, the reinforcement learning objective is defined as the expectation of the advantage-weighted log-likelihood of the sampled outputs:

$$L_{\mathrm{RL}}(\theta) \;=\; \mathbb{E}_{\tau \sim \mathcal{C}} \left[ \frac{1}{G} \sum_{i=1}^{G} A^i \, \log \pi_\theta\big(o_{pos}^i \,\big|\, \tau_{<pos}\big) \;-\; \beta \, \mathrm{KL}(\pi_\theta \,\|\, \pi_{\mathrm{ref}}) \right], \tag{7}$$

where $A^i$ is the group-relative advantage from Eq. 6, $\pi_\theta(o_{pos}^i \,|\, \tau_{<pos})$ is the policy model's probability of generating response $o_{pos}^i$ given the input sequence $\tau_{<pos}$, $\beta$ controls the strength of the KL-divergence penalty, and $\pi_{\mathrm{ref}}$ is the reference model.

**Token-Inclusive Next-Token Prediction** For the majority of tokens with low-entropy that are not exploit for reinforcement learning, we include them for the standard next-token prediction training to acquire comprehensive information. In particular, we first apply masking on the selected tokens that have been used in reinforcement learning to avoid duplicate training. Then, we employ supervised fine-tuning with teacher forcing for next-token prediction. The procedure is as follows.

$$m_{pos} = \mathbb{1}[\tau_{pos} \notin \mathcal{C}], \quad \mathcal{L}_{\mathrm{NTP}}(\theta) = -\sum^{S} m_{pos} \log p_\theta(\tau_{pos} \mid \tau_{<pos}), \tag{8}$$

where $m_{pos}$ is a binary mask specifying whether the token at position $pos$ contributes to the NTP loss, the function $\mathbb{1}[\tau_{pos} \notin \mathcal{C}]$ is the indicator function, equal to 1 if $\tau_{pos}$ is not in $\mathcal{C}$ and 0 otherwise, $\mathcal{L}_{\mathrm{NTP}}(\theta)$ is the next-token prediction loss, and $S$ is the training sequence. To integrate both supervised and reinforcement learning, we combine the token-inclusive next-token prediction loss with the reinforcement loss obtained in Eq. 7:

$$\mathcal{L} = \mathcal{L}_{\mathrm{RL}}(\theta) + \lambda \cdot \mathcal{L}_{\mathrm{NTP}}(\theta), \tag{9}$$

where $\lambda$ is a trade-off weight for balancing $\mathcal{L}_{\mathrm{NTP}}$ and $\mathcal{L}_{\mathrm{RL}}$. This design creates complementary optimization dynamics that leverage the strengths of both paradigms: reinforcement learning improves model non-trivial reasoning decisions via uncertain high-entropy tokens, while next-token prediction handles language understanding with broad coverage of vocabulary.

## 5 EXPERIMENTS

### 5.1 EXPERIMENTAL SETUP

**Datasets and Baselines.** For language modeling, we follow the settings of RPT to employ Omni-MATH dataset and identify token entropy, resulting in a collection of 4051 training data and 200 evaluation data. For baselines, we compare against strong reasoning models R1-Distill-Qwen-14B (denoted as R1-Distill-14B), Qwen3-14B, and the state-of-the-art method RPT. We conduct the experiments in both settings of next token prediction and next token reasoning, with accuracy as the evaluation metric. For continual post-training, we evaluate and post-train the language modeling checkpoints on Skywork dataset (He et al., 2025), which comprises a diverse collection of mathematical problems spanning multiple levels of difficulty. By uniformly stratifying and sampling data with difficulties ranging from 1 to 13, we include 4096 data points as the training set and 600 for evaluation, ensuring balanced coverage across difficulty levels.

**Implementation Details.** For the proposed model, we set the batch size to 128, epochs to 10, learning rate to 1e-6, initial budget $B_0$ to 800, minimal budget $B_{min}$ to 1, decay factor $\gamma$ to 0.2, transition points $t_1$ and $t_2$ to 30% and 70% of the total training steps, the number of rollout G to 8, and trade-off weight $\lambda$ to 0.1. We set max response length to 2048 for language modeling and 4096 for continual post-training. We run all experiments on sixteen NVIDIA Tesla H100 GPUs with 80GB RAM.

## 5.2 Language Modeling

A language modeling task involves training a model to predict the next token in a sequence given its preceding context. We present results in Table 1, reporting performance across tokens of varying difficulty levels, including easy, medium, and hard. We consider two settings for baselines, including next token prediction (NTP) with standard autoregressive approach and next token reasoning (NTR), where a chain-of-thought is generated before the prediction. We denote the proposed models as RMT-R1 that uses R1-Distill-14B as the base model, and RMT-Q3 that depends on Qwen3-14B.

According to the table, NTP methods yield decent results, confirming the effectiveness of pre-trained models and establishing a reliable benchmark. In contrast, NTR methods perform poorly across all difficulty levels, reflecting the absence of explicit reasoning during pre-training stage and the resulting inability to predict tokens with explicit reasoning chains. The current state-of-the-art method RPT demonstrates the potential of reinforcement learning, but does not substantially outperform the base model. By comparison, our method achieves state-of-the-art performance, surpassing the strongest baseline RPT by an average improvement of **+30.74%** with RMT-R1 and **+64.91%** with RMT-Q3. Moreover, results show that stronger base model amplify these gains: RMT-Q3 consistently outperforms RMT-R1 across all difficulty levels. This demonstrates the superiority of our method in harnessing reasoning capabilities in predicting tokens and language modeling.

Table 1: Performance comparison of language modeling. The best and second-best results are highlighted in bold and underlined, respectively.

| Method | Easy | Medium | Hard | Average |
|---|---|---|---|---|
| R1-Distill-14B (NTP) | 42.04 | 31.71 | 19.64 | 31.13 |
| Qwen3-14B (NTP) | 47.89 | 34.61 | 25.01 | 35.84 |
| R1-Distill-14B (NTR) | 4.76 | 2.43 | 2.09 | 3.09 |
| Qwen3-14B (NTR) | 8.47 | 5.08 | 4.51 | 6.02 |
| RPT | 48.67 | 35.84 | 23.03 | 35.85 |
| RMT-R1 | 62.50 | 43.96 | 34.14 | 46.87 |
| Improvement | +28.42% | +22.66% | +48.24% | +30.74% |
| RMT-Q3 | **76.92** | **55.79** | **44.64** | **59.12** |
| Improvement | +58.04% | +55.66% | +93.83% | +64.91% |

## 5.3 Continual Post-Training

To verify the transferability of our approach, we leverage the checkpoints obtained after reinforcement mid-training in Section 5.2 and further post-train them on a distinct dataset Skywork. For fair comparison, we adopt GRPO as the training algorithm. The results are shown in Table 2.

Before post-training, all baseline models exhibit comparable performance, with Qwen3-14B lagging significantly behind the others. However, after undergoing continual post-training, all models show substantial performance gains, with accuracy rising from the low 20s to nearly 50 or higher. Crucially, the proposed method outperforms all

Table 2: Performance comparison before and after post-training.

| Model | Before | After |
|---|---|---|
| R1-Distill-14B | 23.00 | 51.00 |
| Qwen3-14B | 12.84 | 54.17 |
| RPT | 23.17 | 48.50 |
| RMT-R1 | 24.08 | 54.67 |
| RMT-Q3 | **25.17** | **64.33** |
| Improvement | +8.63% | +18.76% |

baselines and delivers the highest gains across both phases. Among all methods, RMT-Q3 achieves a remarkable jump from 25.17 to 64.33 accuracy, showing that checkpoints obtained after reinforcement mid-training can be effectively post-trained to achieve strong downstream performance. Compared to the strongest baseline, RMT-Q3 shows an improvement of **+8.63%** over RPT before post-training, and **+18.76%** over Qwen3-14B after post-training. These results clearly demonstrate the superior transferability and generalization ability of our approach towards post-training stage.

## 5.4 Response Length and Generation Time Analysis

To evaluate the computational efficiency of our approach, we analyze both response length and generation time across the language modeling and continual post-training stages. Here we define

response length as the average number of tokens generated per response, and generation time as the average time required to generate all responses within a batch.

**Superior efficiency during language modeling.** As shown in Table 3, both RMT-R1 and RMT-Q3 generate significantly shorter responses compared to all baselines with 188 and 186 tokens, respectively. This reduces response length to just 21% of RPT (872 tokens) and 12% of Qwen3-14B (1577 tokens). Figure 3 (a) further illustrates this efficiency gain: as training progresses, both response length and generation time for our method decrease substantially. Importantly, this dramatic reduction in computation cost is achieved without sacrificing quality, as our methods maintain superior performance according to Table 1.

Table 3: Response length comparison.

| Method | Length |
|---|---|
| R1-Distill-14B | 1305 |
| Qwen3-14B | 1577 |
| RPT | 872 |
| RMT-R1 | 188 |
| RMT-Q3 | **186** |

**Dynamic adaptation during continual post-training.** According to Figure 3 (b), our method generates substantially shorter reasoning chains compared to RPT at step 0. However, as post-training progresses and the model encounters more complex downstream tasks, it adaptively increases response length to accommodate deeper reasoning requirements. Notably, even at this increased length, our model remains more efficient than RPT while achieving superior performance in Table 3. The generation time curves mirror this adaptive behavior, consistently demonstrating computational efficiency throughout the training process. These results show that our method effectively prevents verbose reasoning without sacrificing accuracy.

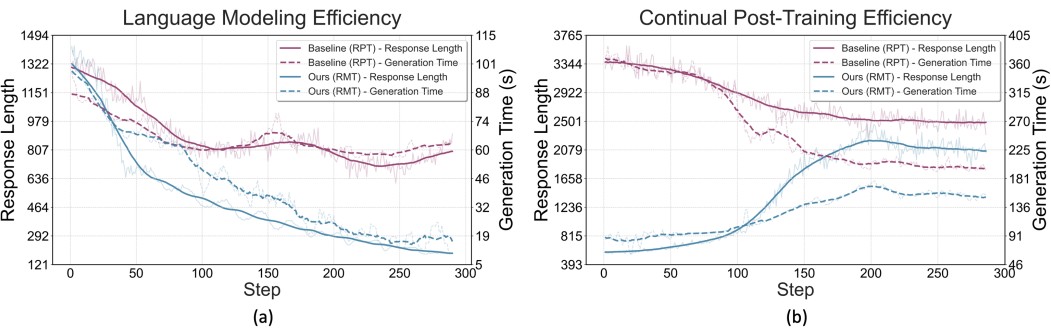

Figure 3: Efficiency comparison in language modeling (a) and continual post-training (b).

## 5.5 ABLATION STUDY

Since RMT contains various designs (i.e., dynamic token budget mechanism (DTB), curriculum-based adaptive sampling (CAS), and dual training strategy with both RL and NTP objectives), we conduct ablation studies to analyze the contributions of different components by removing each of them independently (see Table 4). Specifically, removing the NTP loss significantly affects the performance, highlighting the critical role of next-token prediction in language modeling and the necessity of incorporating comprehensive token information. In addition, the decreasing performances of removing

Table 4: Ablation study.

| Method | Easy | Medium | Hard | Average |
|---|---|---|---|---|
| w/o DTB | 60.35 | 32.79 | 27.59 | 40.24 |
| w/o CAS | 59.32 | 42.66 | 27.68 | 43.22 |
| w/o NTP | 40.00 | 34.26 | 27.27 | 33.84 |
| RMT-R1 | **62.50** | **43.96** | **34.14** | **46.87** |
| w/o DTB | 60.87 | 50.48 | 34.69 | 48.68 |
| w/o CAS | 62.71 | 50.28 | 30.51 | 47.83 |
| w/o NTP | 52.17 | 45.63 | 29.41 | 42.40 |
| RMT-Q3 | **76.92** | **55.79** | **44.64** | **59.12** |

DTB and CAS demonstrate their effectiveness in enhancing the model, while simultaneously improving efficiency. Overall, RMT consistently achieves the best results across all settings, demonstrating that each component contributes meaningfully and that their integration provides a synergistic effect, leading to superior performance.

## 5.6 CASE STUDY

For a more intuitive understanding and comparison, we randomly select an example from OmniMath and present the model generation in Table 5. According to the Table, RMT-Q3 correctly predicts the next token 'probability'. Its reasoning is concise (165 tokens), grammatically correct, and remains

tightly aligned with the question. The model effectively links the incomplete clause to the noun "probability" and avoids unnecessary detours. This focused reasoning connects the context to the intended continuation and produces the correct answer. On the other hand, RPT generates a much longer and less coherent reasoning (1,376 tokens). Its output repeatedly cycles through candidate options and revise itself multiple times. This back-and-forth deviates from the intended semantics, ultimately producing the incorrect prediction "same". Overall, our method is both accurate and concise. It makes the correct decision with a compact and focused reasoning chain, while RPT is verbose, inconsistent in its guesses, and returns the incorrect answer.

Table 5: Case study. The proposed method produces concise and coherent reasoning traces, whereas existing method often generate verbose and inconsistent reasoning, characterized by frequent hesitations and shifts. Key words and sequences that guide the reasoning process are highlighted in gray.

---

**Input:** *Complete the given text by predicting the next token, list multiple potential tokens and select the most probable one as the final prediction. The text is: "Question: Six students sit in a row and leave at random times. What is the probability that one student must pass another to exit the row? Answer: The probability $p$ that no student will have to pass by another student to exit the row is the "*
**Expected Answer:** Probability

---

**RMT-Q3**
**Reasoning:** Hmm, so I need to predict the next token after the given text. First, I'll look at the structure of the last sentence: ... The most probable next word is "probability" since the context is about probability, and the sentence structure suggests it should continue the idea introduced before. Looking at the tokens, the next word should follow grammatically. ... Given the context, "probability" fits perfectly here. So, the next token is most likely "probability" as it follows the context and grammatical structure.
**Answer:** Probability
**Length:** 165

---

**RPT**
**Reasoning:** Okay, so I need to predict the next token in the given text. Let's see, the current text is ... The question is about ... The sentence is incomplete. The structure after "is the" might be followed by a word like "probability", "chance", "number", or maybe "case". Alternatively, perhaps it's "probability $p$... is the..." so maybe the word that comes after "is the" would be something like "probability", but wait, the sentence already starts with "The probability $p$...", so maybe it's a fraction or something. Wait no, maybe the word after is "probability" itself, but that seems redundant. Alternatively, perhaps the sentence is saying it's the "number", so maybe the next word is "number". Or maybe it's the reciprocal or some function. Wait, let me think. The question is ... So maybe the next word after "is the" would be "same" as some known probability, but perhaps in the text it's phrased as "the probability... is the reciprocal...", but maybe not. Alternatively, perhaps the word after is "case", as in "the probability... is the case", but that might not make sense. Alternatively, maybe the word is "same" as another scenario. Wait, maybe in the original text, it's phrased as ... Alternatively, perhaps the text is "the probability $p$... is the number of ways...", so maybe the word after is "number". Alternatively, maybe it's "the probability $p$ is the reciprocal...", so maybe "reciprocal". But looking back, ... But without more context, perhaps the most probable word after "is the" is "same", because maybe the probability p is the same as some other scenario. ... So, based on this, I think the most probable token after "is the" is "same". Wait, but perhaps "same" isn't the correct word. Alternatively, maybe it's "case", "value", "chance", etc. Hmm. Alternatively, ... But given that the question is asking for the probability p that no student will have to pass another, perhaps the text continues as "the probability $p$... is the same as...", so "same" would come next.
**Answer:** same
**Length:** 1376

---

## 6    CONCLUSION

In this paper, we formally introduce and define the problem of Reinforcement Mid-Training, an intermediate stage that strengthens large language models before downstream post-training. We identify three key challenges in this setting: excessive computational overhead from overthinking, learning ineffectiveness caused by imbalanced token entropy, and underutilization of token information. To address these challenges, we propose RMT, a reinforcement mid-training framework that is efficient, adaptive, and unified. Extensive experiments show that RMT improves both performance and efficiency, producing models better prepared for subsequent post-training stage. This work positions Reinforcement Mid-Training as a valuable new stage in the LLM lifecycle and opens up promising directions for future research.

## 7 ACKNOWLEDGMENT

This work was supported in part by the National Natural Science Foundation of China, under Grant 62441227.

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
