# OpenReview forum: "Reinforcement Mid-Training"
_ICLR.cc/2026/Conference — ICLR 2026 Poster_

### Official Review · Reviewer_gdXM · 2025-10-16

**Soundness:** 2
**Presentation:** 2
**Contribution:** 2
**Rating:** 4
**Confidence:** 4

**Summary:**

This paper introduces Reinforcement Mid-Training (RMT), a distinct stage in the LLM development positioned between pre-training and post-training. The proposed RMT framework addresses key challenges of inefficiency, imbalanced data difficulty, and information underutilization by integrating a dynamic token budget, a curriculum-based adaptive sampling strategy, and a dual training objective. The work demonstrates improvements in both performance and computational efficiency on mathematical reasoning tasks, establishing a foundation for subsequent post-training.

**Strengths:**

- The paper clearly identifies three practical challenges (i.e., overthinking, entropy imbalance, token underutilization) that are highly relevant to current research.
- The proposed RMT framework is a well-designed solution where each component (DTB, CAS, dual training) directly addresses one of the identified challenges.
- The empirical results are convincing, showing substantial gains not only in accuracy but also in efficiency.

**Weaknesses:**

- My main concerns lie in the paper's foundational assumptions about *difficulty* and *efficiency*. The framework relies heavily on token entropy as a proxy for reasoning difficulty, which is a significant simplification. High uncertainty does not always correlate with high cognitive complexity. Similarly, the dynamic token budget enforces a *shorter is better* heuristic, which may not hold for problems that intrinsically require elaborate, long-chain reasoning. These simplified proxies could limit the method's robustness and generalizability, potentially causing it to penalize valid, complex reasoning paths in favor of concise but possibly suboptimal solutions.
- The experimental validation is predominantly confined to the domain of mathematical reasoning. This narrow focus makes it difficult to assess the framework's effectiveness on other complex reasoning tasks like code generation, legal analysis, or creative problem-solving.
- The framework introduces a considerable number of new hyperparameters (e.g., budget decay, curriculum transition points, trade-off weights), but there lacks a sensitivity analysis, leaving its practical robustness and ease of implementation in question.

**Questions:**

- How would the performance of the curriculum learning component change if difficulty were defined by a more semantically aware metric (e.g., reasoning step complexity) instead of token entropy?
- Could the dynamic token budget be made adaptive to the input's complexity rather than decaying with the training step, for instance, by allocating a larger initial budget for demonstrably harder problems?

---

> ### Author Response · Authors · 2025-11-22
> **Response to Reviewer gdXM (1/2)**
>
> Dear Reviewer gdXM,
>
> We appreciate your comments. We make clarifications to your questions and concern as follows. If our response can address your concerns, we really hope you can increase your score. Thank you!
>
> **Q1**: High uncertainty does not always correlate with high cognitive complexity. The dynamic token budget may penalize valid, complex reasoning paths.
>
> **A1**:
> Thank you for the question. We agree that more advanced methods for estimating difficulty can be valuable. In our work, we use token entropy as a proxy because it is simple and intuitive. Our approach is grounded in the following four key observations:
>
> 1. Latest research support: We follow the findings of the latest NeruIPS 2025 paper [1] to treat high-entropy tokens as indicators of challenging reasoning steps.
> 2.	Broad coverage but efficient token selection: We sample a small set of tokens for RL training in proportion to their entropy, which enables an effective balance between training efficiency and broad coverage.
> 3.	Token budget as a reward mechanism: Rather than truncating model outputs, we incorporate the token budget directly into the reward function.
> 4.	Flexible inference-time behavior: During inference, the model is permitted to use as many tokens as it deems necessary when additional elaboration is beneficial.
>
> In particular, paper [1] states that high-entropy minority tokens contribute the majority of RLVR gains, whereas low-entropy majority tokens offer limited benefit and can even be detrimental. Consequently, we focus RL updates on a small subset of high-entropy next tokens, achieving substantial improvements with significantly reduced compute. Empirically, model accuracy is higher on “easy” (low-entropy) tokens and lower on “hard” (high-entropy) tokens, reinforcing entropy as an effective proxy for reasoning difficulty.
>
> To improve efficiency and guide behavior, we incorporate a dynamic token budget directly into the reward function. This mechanism encourages concise and effective chains of thought without truncating outputs, leading the model to adopt more efficient reasoning strategies. Ablation results (Table 4 in the paper) indicate that removing the budget both reduces accuracy and increases redundancy.
>
> [1] Wang, S., et al. Beyond the 80/20 Rule: High-Entropy Minority Tokens Drive Effective Reinforcement Learning for LLM Reasoning. NeurIPS 2025.
>
>
> **Q2**:The experiment is confined to the domain of mathematical reasoning.
>
> **A2**: Thanks for your suggestion. We conduct additiona experiments on 93 distinct domains using the dataset GeneralThoughtArchive. We find that our method RMT still show significant performance improvement (ranging from +18.63% to +64.52%) compared to existing SOTA methods. The experiment results and details are shown as follows.
> | Model                | Easy                 | Medium               | Hard                 | Average              |
> | -------------------- | -------------------- | -------------------- | -------------------- | -------------------- |
> | Base (R1-Distill-14B) | 3.25                 | 1.27                 | 1.94                 | 2.15                 |
> | RPT (SOTA)           | 31.58                | 22.38                | 11.98                | 21.98                |
> | RMT (ours)           | **40.00** | **26.55** | **19.71**  | **28.75** |
> | Improvement        | +26.67% | +18.63% | +64.52% | +30.80% |
>
> In particular, GeneralThoughtArchive dataset covers 93 distinct domains, such as humanities (literature, history, philosophy), natural sciences (physics, chemistry, biology), and code generation. Given the time constraint, we performed stratified sampling based on task types to select 9,000 samples for training and 1,000 samples for testing. We compare the performance between our proposed RMT and the current SOTA method RPT method, with the base model r1-distill-14B model.

---

> > ### Author Response · Authors · 2025-11-22
> > **Response to Reviewer gdXM (2/2)**
> >
> > **Q3**: The framework lacks a sensitivity analysis.
> >
> > **A3**: Thank you for your suggestion. We conduct additional experiments on parameter sensitivity, including curriculum transition points, trade-off weight, and the token budget. We summarize our findings up front: (1) Our method is robust to reasonable choices of curriculum transition points and the trade-off weight $ \lambda $, yielding stable average performance. (2) A reasonable small token budget can be sufficient to achieve decent performance. Reducing the budget further can lead to degradation. The details and results for each experiment are shown below.
> >
> > **A3.1: Curriculum Transition Points**
> >
> > We denote transition ratios $tr_1=t_1/T$ and $tr_2=t_2/T$. We vary curriculum transition points around the paper’s setting (0.3, 0.7) and observe that the overall performance remains stable, and spending more time on extremes improves those extremes. In other words, if more training resource is allocated to the easy tokens, the model will have a better performance on easy tokens. The setting adopted in our paper provides a balanced trade-off.
> >
> > | Parameters                         | Easy  | Medium | Hard  | Average |
> > |-----------------------------------|-------|--------|-------|---------|
> > | $tr_1=0.1, tr_2=0.5$            | 50.00 | 46.80  | **38.64** | 45.15   |
> > | $tr_1=0.3, tr_2=0.7$ (Paper)    | 62.50 | 43.96  | 34.14 | **46.87**   |
> > | $tr_1=0.5, tr_2=0.9$            | **66.67** | **48.62**  | 23.00 | 46.09   |
> >
> > **A3.2: Trade-off Weight ($ \lambda $)**
> >
> > We select $ \lambda \in \\{0.05, 0.1, 0.2\\} $ and compare the performance. We observe that setting $ \lambda$ to a small value of 0.05 harms overall performance, while setting it to a large value of 0.2 impairs learning on hard tokens. While setting $ \lambda=0.1 $ is a good balance, we acknowledge that $ \lambda=0.2 $ can even achieve a better performance than what has been reported in the paper.
> >
> > | Parameters                | Easy  | Medium | Hard  | Average |
> > | ------------------------- | ----- | ------ | ----- | ------- |
> > | $ \lambda=0.05 $        | 59.26 | 44.21  | 25.46 | 42.97   |
> > | $ \lambda=0.1 $ (Paper) | 62.50 | 43.96  | **34.14** | 46.87   |
> > | $ \lambda=0.2 $         | **68.18** | **45.28**  | 28.57 | **47.34**   |
> >
> >
> > **A3.3: Token Budget**
> >
> > We vary the initial token budget while keeping the token budget decay rate the same. Results show that using an insufficient token budget can significantly degrade the performance. Once the budget is sufficient, further increasing it can also have impact to the model performance.
> >
> > | Parameters                                  | Easy  | Medium | Hard  | Average |
> > |--------------------------------------------|-------|--------|-------|---------|
> > | budget=500        | 43.24 | 34.78  | 17.39 | 31.80   |
> > | budget=800 (Paper)| **62.50** | **43.96**  | 34.14 | **46.87**   |
> > | budget=1500       | 60.71 | 33.65  | **35.71** | 43.35   |
> >
> >
> > **Q4**: Using a more semantically aware metric instead of token entropy for difficulty of curriculum learning component.
> >
> > **A4**: Thank you for the suggestion. We use token entropy to guide curriculum learning primarily due to its simplicity and efficiency. We acknowledge that introducing more advanced metrics such as sequence-level or reasoning‑step complexity have the potential to further boost performance. However, this usually introduces additional cost as they can require step‑level annotations or additional inference passes to estimate per‑example complexity. In contrast, token entropy is obtained directly from a single forward pass, allowing us to leverage a useful difficulty signal without introducing additional computational overhead.
> >
> > **Q5**: Make token budget adaptive to the input's complexity rather than decaying with the training step.
> >
> > **A6**:  Thank for your suggestion. We agree that the token budget can be made input‑adaptive. Practically, one can maintain a modest floor budget and grant additional tokens to examples flagged as hard by inexpensive online signals (e.g., high early‑step entropy/loss, repeated failures in prior inference, or large variance across shallow self‑consistency samples). This enables the allocation of a larger initial budget to demonstrably harder problems. We leave the exploration of this direction to future work.
> >
> > Please let us know if there are any remaining concerns that have not been addressed. We appreciate any feedback.

---

### Official Review · Reviewer_Vakx · 2025-10-19

**Soundness:** 3
**Presentation:** 3
**Contribution:** 3
**Rating:** 6
**Confidence:** 4

**Summary:**

This paper proposes **RMT**, a “reinforcement mid-training” stage placed between pre-training and post-training. RMT targets three issues—**overthinking**, **imbalanced token-entropy**, and **wasted token information**—via: (1) a **dynamic token budget** with a triangular **length reward** to keep chain-of-thought concise, (2) an **entropy-based curriculum** that shifts sampling from easy→hard tokens over time, and (3) a **dual objective** that runs **RL on selected high-entropy tokens** while using **next-token prediction (NTP)** on the rest. On OmniMATH language-modeling, RMT reportedly improves accuracy by **up to +64.91%** while using **≈21%** of the reasoning length vs. baselines, and the mid-trained checkpoints further boost later post-training by **up to +18.76%** on Skywork.

**Strengths:**

- The paper clearly defines **reinforcement mid-training** and motivates why an intermediate stage can help later post-training.
- The **dynamic token budget** with a **triangular length reward** is an intuitive way to curb overthinking without eliminating reasoning.
- The **entropy-based curriculum** is straightforward to implement and aligns with progressive difficulty.

**Weaknesses:**

- The **evaluation scope is narrow**, focusing on math-reasoning (OmniMATH/Skywork) with only **200 evaluation samples** in the LM setup, which limits statistical confidence and domain generality.
- The **reward is exact next-token match** (Eq. 5), which may bias learning toward surface-form correctness rather than semantic adequacy; the paper does not compare against richer RL signals (e.g., verifier or sequence-level rewards).
- Several **strong dynamic RL baselines** are missing in experiments (e.g., recent efficient RL variants or learned curricula), even though related work mentions them; this makes the SOTA claim harder to verify.

**Questions:**

Have you tried **sequence-level** or **verifier-based** rewards (beyond exact token match) and, if so, how do they change the efficiency/accuracy trade-off?

---

> ### Author Response · Authors · 2025-11-21
> **Response to Reviewer Vakx**
>
> Dear Reviewer Vakx,
>
> We really appreciate your comments and suggestions. We hope our response can address your questions. Please find our detailed response below.
>
> **Q1**:The evaluation only focuses on math domain.
>
> **A1**: Thanks for your suggestion. We conduct additiona experiments on 93 distinct domains using the dataset GeneralThoughtArchive. We find that our method RMT still show significant performance improvement (ranging from +18.63% to +64.52%) compared to existing SOTA methods. The experiment results and details are shown as follows.
> | Model                | Easy                 | Medium               | Hard                 | Average              |
> | -------------------- | -------------------- | -------------------- | -------------------- | -------------------- |
> | Base (R1-Distill-14B) | 3.25                 | 1.27                 | 1.94                 | 2.15                 |
> | RPT (SOTA)           | 31.58                | 22.38                | 11.98                | 21.98                |
> | RMT (ours)           | **40.00** | **26.55** | **19.71**  | **28.75** |
> | Improvement        | +26.67% | +18.63% | +64.52% | +30.80% |
>
> In particular, GeneralThoughtArchive dataset covers 93 distinct domains, such as humanities (literature, history, philosophy), natural sciences (physics, chemistry, biology), and code generation. Given the time constraint, we performed stratified sampling based on task types to select 9,000 samples for training and 1,000 samples for testing. We compare the performance between our proposed RMT and the current SOTA method RPT method, with the base model r1-distill-14B model.
>
>
> **Q2**：Changing next-token reward to sequence-level or verifier-based rewards?
>
> **A2**：Thanks for the suggestion. For the sake of not introducing additional complexities, we did not try verifier-based reward or sequence-level reward. Our paper focus on the most fundamental scenario of unsupervised learning from unlabeled data. Thus, we leverage the next token as token level reward, as in llm pre-training stage. Using sequence-level or verifier-based rewards introduce unnessary complexities as it either requires multiple tokens or an additional trained verifier. Moreover, token-level reward already show significant performance improvement.
>
> **Q3**: Several strong dynamic RL baselines are missing in experiments (e.g., recent efficient RL variants or learned curricula), even though related work mentions them; this makes the SOTA claim harder to verify.
>
> **A3**: We are not proposing a new RL algorithm. For a fair comparison, we adopt the same RL algorithm (GRPO) as the baseline. In other words, GRPO is just a method we used to evaluate on top of our innovations, i.e., a dynamic token budget mechanism, a curriculum-based adaptive sampling method, and a dual training strategy. Our paper focus on addressing the challenges inherent in applying whatever RL algorithms, such as entropy imbalance and token wastage caused by overthinking.
>
> However, we acknowledge that adopting a better RL algorithm than GRPO may further improve the performance. To further validate, we conduct additional experiments using a more advanced RL algorithm (DAPO) rather than GRPO. The results are shown in the following table. We can conclude that even with a more advanced post-training RL algorithm, our proposed method can still outperform the existing SOTA method RPT with a great margin.
> | Model                | Before                 | After               |
> | -------------------- | -------------------- | -------------------- |
> RPT (SOTA) |  23.17 | 52.30 |
> RMT (ours) |   **24.08** |  **57.83** |
> Improvement | +3.93% | +10.57% |
>
>
> Please let us know if there are any remaining concerns that have not been addressed. We appreciate any feedback.

---

### Official Review · Reviewer_D7Vg · 2025-11-03

**Soundness:** 3
**Presentation:** 3
**Contribution:** 3
**Rating:** 6
**Confidence:** 4

**Summary:**

This paper introduces RMT, a framework for efficient, adaptive, and unified reinforcement mid-training. It proposes several strategies to enhance both training efficiency and performance. These include: 1) a dynamic token budget mechanism, which prevents the generation of excessively long chains of thought that can impede training efficiency; 2) a curriculum learning strategy that determines token difficulty based on entropy, allowing the model to gradually progress from simpler to more complex tokens during training; and 3) a combined training approach that utilizes both cross-entropy and Reinforcement Learning (RL) loss. The experimental results demonstrate that RMT improves next-token prediction capabilities and generalizes effectively to subsequent post-training with RL.

**Strengths:**

1. The method shows strong performance, significantly enhancing the model's ability for next-token prediction and demonstrating good generalization to the post-training RL phase.
2. The paper is well-written, and the proposed method is straightforward to understand.

**Weaknesses:**

1. I was unable to find some specific experimental details. For instance, what proportion of tokens were subjected to reinforcement learning training versus the standard cross-entropy objective?

2. I am curious about the added value of reinforcement mid-training in a scenario where the post-training RL phase is exceptionally thorough and prolonged. Would the improvements from RMT still be significant if the model undergoes a very comprehensive post-training RL alignment?

**Questions:**

Could the authors elaborate on the initial stages of training? Was a "warm-up" phase or a specific set of prompts used to enable the model to learn how to reason for the next token from a pre-trained model?

---

> ### Author Response · Authors · 2025-11-21
> **Response to Reviewer D7Vg**
>
> Dear Reviewer D7Vg,
>
> Thank you very much for your comments. We make clarifications to your questions as follows. If our response can address your questions, we really hope you can increase your score. Thank you!
>
> **Q1**: What is the proportion of tokens subjected to RL training?
>
> **A1**: To maintain an efficient training pipeline, we randomly sample a small subset of tokens from each data instance during RL training, since RL is substantially more time-consuming than next-token prediction. In particular, we randomly select one easy token, two medium tokens, and one hard token from all tokens of a data sample during RL training. This allows us to balance efficiency with broad coverage. We want to highlight that although we are using a small subset of tokens, we observe significant performance improvements, as shown in Table 1.
>
>
> **Q2**: Significance of RMT when facing a very comprehensive post-training.
>
> **A2**: Thank you for asking. To demonstrate the effectiveness of RMT when facing a more comprehensive post-training, we are running additional experiments using a more advanced RL algorithm (DAPO) rather than GRPO. The results are shown in the following table. We can conclude that even with a more advanced post-training RL algorithm, our proposed method can still outperform the existing SOTA method RPT with a great margin.
> | Model                | Before                 | After               |
> | -------------------- | -------------------- | -------------------- |
> RPT (SOTA) |  23.17 | 52.30 |
> RMT (ours) |   **24.08** |  **57.83** |
> Improvement | +3.93% | +10.57% |
>
> In addition, we want to reiterate that our existing experiments answers the question. As shown in Table 2,  RMT can significantly improves the upper bound of post-training performance, by comparing row 1 and row 4 for R1 base model, and comparing omparing row 2 and row 5 for Qwen3 base model. In addition, RMT checkpoint with post-training (row "After") yields consistently higher scores than before post-training (row "Before").
>
> **Q3**：Did the authors use a warm-up phase to enable next token prediction?
>
> **A3**: Thank you for your suggestion. We did not incorporate a warm-up phase, as we found that guiding the model with a prompt alone was sufficient to achieve a deent performance in next-token prediction. However, we acknowledge that a warm-up stage may offer additional performance gains. The prompt template we used is adopted from [1] with some minor modifications. We observed that the model can successfully follow the requirement when generating answers. The prompt is provided as follows:
>
> ```
> Complete the given text wrapped in ```and ```by predicting the next token, list multiple potential tokens and select the most probable one as the final prediction. Wrap your final prediction in \\boxed{} (note: the token may begin with a space, e.g., \\boxed{ para} or \\boxed{ =}; do not use \\text{}). The context is: ```{CONTEXT}```, now please predict the next token.
>
> IMPORTANT: In your thought process, that is, between the "<think>" and "<\think>" tags, you should use exactly {token_budget} tokens.
> ```
>
> [1] Dong, Qingxiu, et al. "Reinforcement Pre-Training." arXiv preprint arXiv:2506.08007 (2025).
>
> Please let us know if there are any remaining concerns that have not been addressed. We appreciate any feedback.

---

> > ### Comment · Reviewer_D7Vg · 2025-11-27
> >
> > Thank you for the response, which resolve my concerns.

---

### Author Response · Authors · 2025-11-22
**General Response to Reviewers**

Dear All Reviewers:

We appreciate all of your feedback and constructive suggestions. We are pleased that you acknowledged the contributions of our work: “The method shows strong performance, significantly enhancing the model's ability for next-token prediction and demonstrating good generalization to the post-training RL phase.” (Reviewer D7Vg); “The paper clearly defines reinforcement mid-training and motivates why an intermediate stage can help later post-training.” (Reviewer Vakx); “The proposed RMT framework is a well-designed solution where each component (DTB, CAS, dual training) directly addresses one of the identified challenges. The empirical results are convincing, showing substantial gains not only in accuracy but also in efficiency.” (Reviewer gdXM).

Many questions are related to method details (Q1, Q3 of Reviewer D7Vg, Q2 of Reviewer Vakx, Q1, Q4, Q5 of Reviewer gdXM) and experiment setting (Q2 of Reviewer D7Vg, Q1, Q3 of Reviewer Vakx, Q2, Q3 of Reviewer gdXM), which we have addressed in our detailed responses.

We further conduct additional experiments to answer the questions and support our claims:

- **Generalization to broader domains:** Evaluating the performance on 93 distinct domains using the GeneralThoughtArchive dataset to demonstrate the method's effectiveness beyond mathematical reasoning (Q1 of Reviewer Vakx, Q2 of Reviewer gdXM).

- **Parameter sensitivity analysis:** Conducting comprehensive sensitivity tests to understand the stability of the model, including curriculum transition points, the trade-off weight $\lambda$, and the initial token budget (Q3 of Reviewer gdXM).

- **Extension of our method to more RL algorithms:** Validating the performance of RMT with the DAPO algorithm.  This extension enables us to evaluate whether RMT continues to provide stable improvements under stronger post-training regimes, and see if the performance can be further improved (Q2 of Reviewer D7Vg, Q3 of Reviewer Vakx).

We sincerely hope our response can address your concerns, and we would be grateful if you could consider increasing your rating. Please let us know if you have any remaining questions or comments. We will happily respond. Thank you!

---

### Meta-Review · Area_Chair_8e3P · 2025-12-16

**Summary:**

The paper formalizes reinforcement mid-training (RMT) as a distinct stage between pre-training and post-training for LLMs and proposes three components: a dynamic token budget with a triangular length reward, an entropy-based curriculum sampling scheme, and a dual objective that applies RL to selected high-entropy tokens while using next-token prediction (NTP) on the remaining tokens. Empirical results on OmniMATH and transfer to Skywork show sizable accuracy and efficiency gains, with ablations supporting the contribution of each component. The rebuttal addresses several key weaknesses raised by the reviewers by providing multi-domain experiments (GeneralThoughtArchive, 93 domains), sensitivity analyses for major hyperparameters, details on token selection for RL, and additional baselines using a stronger RL algorithm (DAPO). Remaining concerns include the reliance on token entropy as a proxy for difficulty and the use of exact next-token match rewards, which may limit semantic robustness and generality. Overall, the work is novel and methodologically sound enough for acceptance as a poster, with clear impact on training efficiency and a plausible pathway to better post-training results. The rebuttal did overcome several objections, though some issues remain for future work.

**Reviewer Concerns:**

#### Reviewer_gdXM
1. **Concern**: Entropy-as-difficulty proxy may misalign with cognitive complexity; DTB may penalize valid long reasoning.
   - **Why Unresolved**: Partially addressed. Authors justify entropy with recent evidence and ablations (Table 4) and shape length via reward rather than truncation; inference is unconstrained. Still no empirical comparison with semantically-aware difficulty metrics or input-adaptive budgets beyond a speculative suggestion.
   - **Impact on Decision**: Moderate. Mitigation is credible enough for acceptance, but absence of empirical tests with alternative difficulty signals and adaptive budgets prevents stronger recommendation.

2. **Concern**: Evaluation narrowly on math reasoning; generalizability unclear.
   - **Why Unresolved**: Addressed. Rebuttal adds GeneralThoughtArchive experiments over 93 domains with stratified sampling (9k train / 1k test), showing consistent gains (+18.63% to +64.52%).
   - **Impact on Decision**: Mitigated. Cross-domain evidence supports poster acceptance.

3. **Concern**: No sensitivity analysis despite many hyperparameters.
   - **Why Unresolved**: Addressed. Rebuttal includes sensitivity for transition points (t1,t2), λ, and initial budget, showing relative robustness and trade-offs (A3.1–A3.3).
   - **Impact on Decision**: Mitigated. Improves confidence in practical deployability.

4. **Concern**: Request for input-adaptive token budget and semantically-aware difficulty metric.
   - **Why Unresolved**: Unresolved. Authors acknowledge as future work; no experiments provided.
   - **Impact on Decision**: Minor. Does not block acceptance; limits placement.

---

#### Reviewer_Vakx
1. **Concern**: Small evaluation (200 samples) and math-only focus limit statistical confidence and domain generality.
   - **Why Unresolved**: Partially addressed. The new 93-domain experiment (1k test) increases scope and sample size. The core OmniMATH setup remains small, but added results ease the concern.
   - **Impact on Decision**: Moderate. With broader evidence, acceptable for poster.

2. **Concern**: Exact next-token reward (Eq. 5) may bias learning; lack of sequence-level/verifier rewards comparison.
   - **Why Unresolved**: Unresolved. Authors chose simplicity and did not evaluate richer signals; no verifier/sequence-level reward ablations were provided.
   - **Impact on Decision**: Moderate. Prevents stronger recommendation; suggests follow-up work.

3. **Concern**: Missing dynamic RL baselines or learned curricula beyond GRPO; SOTA claim hard to verify.
   - **Why Unresolved**: Partially addressed. Authors keep GRPO for fairness and add DAPO extension showing gains, but do not compare with other recent efficient RL or curriculum methods mentioned in related work.
   - **Impact on Decision**: Moderate. Comparative scope is improved but still incomplete.

---

#### Reviewer_D7Vg
1. **Concern**: Proportion of tokens trained with RL vs NTP; early training setup (warm-up); impact under thorough post-training.
   - **Why Unresolved**: Resolved. Authors specify sampling (1 easy, 2 medium, 1 hard per instance), describe prompt-guided setup without warm-up, and provide DAPO and Skywork results (Table 2; DAPO table) demonstrating sustained benefits post-training. Reviewer confirms resolution.
   - **Impact on Decision**: Mitigated. Increases confidence; supports acceptance.

**Reviewer Scores:**

#### Reviewer_gdXM
- **Original Score**: 4
- **Expected Score After Discussion**: 4
- **Rationale**: Broader cross-domain evidence and hyperparameter sensitivity strengthen robustness; clarification that DTB is a reward not a hard cap and inference is unconstrained reduces risk of penalizing valid reasoning. However, lack of experiments with semantically-aware difficulty or input-adaptive budgets and absence of richer reward signals remain.

---

#### Reviewer_Vakx
- **Original Score**: 6
- **Expected Score After Discussion**: 6
- **Rationale**: New 93-domain results and DAPO extension improve scope and comparators, but the core reward remains exact token match without alternative RL signals, and strong dynamic RL/curriculum baselines are still missing. The reviewer’s core reservations persist, suggesting a steady score.

---

#### Reviewer_D7Vg
- **Original Score**: 6
- **Expected Score After Discussion**: 6  or 8
- **Rationale**: Methodological clarifications and post-training evidence (DAPO; Table 2) addressed the reviewer’s concerns; the reviewer explicitly acknowledged resolution.

---

### Decision · Program_Chairs · 2026-01-26

Accept (Poster)